# Regulatory Effects Mediated by *Enteromorpha prolifera* Polysaccharide and Its Zn(II) Complex on Hypoglycemic Activity in High-Sugar High-Fat Diet-Fed Mice

**DOI:** 10.3390/foods12152854

**Published:** 2023-07-27

**Authors:** Liyan Li, Yuanyuan Li, Peng Wang

**Affiliations:** 1Medical School, Huanghe Science and Technology College, Zhengzhou 450063, China; liyanli@hhstu.edu.cn; 2Food Science and Engineering College, Ocean University of China, Qingdao 266003, China; yuanyuanli@163.com

**Keywords:** *E. prolifera* polysaccharide, Zn(II) complex, hypoglycemic activity

## Abstract

In order to investigate and develop functional foods of marine origin with hypoglycemic activity, *Enteromorpha prolifera* polysaccharide–Zn(II) (EZ) complex was first prepared by marine resourced *E. prolifera* polysaccharide (EP) and ZnSO_4_ and their anti-diabetes activities against high-sugar and high-fat-induced diabetic mice were evaluated. The detailed structural characterization of EZ was elucidated by UV-Vis spectroscopy, infrared spectroscopy, and monosaccharide composition determination. The pharmacological research suggests that EZ has a potent hypoglycemic effect on high-sugar and high-fat-induced diabetic mice by inhibiting insulin resistance, improving dyslipidemia, decreasing inflammatory status, repairing pancreas damage, as well as activating the IRS/PI3K/AKT signaling pathway and regulating GLUT2 gene expression. At the same time, microbiota analysis indicates that a high dose of EZ could enhance the abundance of dominant species, such as *Staphylococcaceae*, *Planococcaceae*, *Muribaculaceae*, *Aerococcaceae*, and *Lacrobacillaceae*, in intestinal microbiota distribution. Thus, EZ could be considered as a potential candidate for developing an ingredient of functional foods for Zn(II) supplements with hypoglycemic activity.

## 1. Introduction

Type 2 diabetes, also known as non-insulin-dependent diabetes, is a chronic metabolic disease characterized by high blood sugar, relative insulin deficiency, and insulin resistance. It accounts for almost 90% of diabetes cases and mainly occurs in adults who are obese and inactive. The disease can be prevented by maintaining a normal weight, exercising regularly, and eating a proper diet. It can be treated in the early stages by taking exercise, changing diet, and drugs [1], while due to the side effects of pure chemical synthetic diabetic drugs, numerous natural products, such as marine polysaccharides, have been investigated and identified for their antidiabetic activity [2,3].

*E. prolifera* is one kind of green algae and the dominant species of green tides in the Yellow Sea of China [4]; its main component is polysaccharide [5]. Previous studies have shown that the polysaccharide has many biological activities, such as hypoglycemia [6,7,8], hypolipidemic [9,10] and immunomodulatory activity [11,12], antioxidant [13,14], gut microbiota modulation [15], and so on. Polysaccharide has been reported as a forage supplement for chicken, fish, and piglets to improve their growth performance based on its bioactivities [16].

Diabetes is associated with the disruption of zinc homeostasis [17]. Zn(II), namely a bivalent zinc ion, is secreted by pancreatic β cells during insulin secretion. Due to participation in insulin storage and secretion, Zn(II) has absorbed a lot of attention in the development of effective antidiabetic agents. In addition, Zn(II) can improve the diseases associated with diabetic complications, such as nephropathy and cardiomyopathy [18]. Therefore, Zn(II) supplements have been reported to have beneficial effects on diabetic patients [19]. Yoshikawa identified ligands that could improve the bioavailability of the Zn(II) complex and afford benefits on glucose metabolism [20]. Thus, several polysaccharides, as ligands, were combined with Zn(II) to develop effective antidiabetic agents [1,21]; however, *E. prolifera* polysaccharides (EP) originating from marine have not been reported to combine with Zn.

EP was also reported to possess an excellent chelating property [22]; therefore, EP–Zn(II) (EZ) complex was prepared for the first time, and its hypoglycemic activities against high-sugar and high-fat (HSHF)-induced diabetic mice were investigated in the present research. This is an important attempt to combine natural products with zinc to develop functional foods with potent hypoglycemic ability and improve the medicinal value of EP.

## 2. Materials and Methods

### 2.1. Materials

ZnSO_4_ was purchased from Sinopharm Group (China). Monosaccharide standards were purchased from Sigma-Aldrich Co., Ltd. (St. Louis, MO, USA). Total cholesterol (TC) and Triglyceride (TG) assay kits were purchased from Nanjing Jiancheng Bioengineering Institute (Nanjing, China). The ELISA kits were purchased from Solarbio Company (Beijing, China). Other chemicals and reagents were all of analytical grade.

### 2.2. Preparation and Optimization of EZ

EP was extracted as previously reported by Cui [23]. Based on the fixed variables of 40 °C, 60 min, pH 5.0, and EP:ZnSO_4_ at 1:1, analysis of single factor experiments were carried out by changing single variables, including temperatures (30, 40, 50, 60, and 70 °C), reaction times (30, 60, 90, 120, and 150 min), pH values (4.0, 4.5, 5.0, 5.5, and 6.0) and the proportions of EP and ZnSO_4_ (3:1, 2:1, 1:1, 1:2, and 1:3, *w*/*w*), respectively. The complex of EP and ZnSO_4_ was synthesized according to the optimal single factors. After centrifuging at 4500 rpm for 15 min, the supernatant was dialyzed with dialysis membrane of 200 Da, and then freeze dried. The EP–Zn(II) complex was obtained and named as EZ.

### 2.3. Characterization of EZ

The molecular weights of the polysaccharide and its Zn(II) complex were determined according to the method described by Chi et al. [24]. The Zn(II) content of EZ was determined by flame atomic absorption spectrometry described by Dong et al. [22]. The Fourier transform infrared (FI-IR) spectra of EZ were recorded by Magna-IR560 spectrometer (ThermoFisher, Madison, WI, USA) according to the description by Cui et al. [25]. The monosaccharides of EZ were quantified by reversed-phase HPLC after PMP (1-phenyl-3-methyl-5-pyrazolone) precolumn derivatization [26].

### 2.4. Safety Evaluation of EZ

In total, 72 healthy SPF-grade ICR mice (albino strain, weight, 18–22 g; age, 7 weeks old; half male and half female) were purchased from Pengyue Laboratory Animal Technology Co., Ltd. (Jinan, China) with the mouse license of SCXK (LU) 2014-0007. An acute toxicology experiment was carried out according to the method of Zhang Y, et al. [1] in doses of EZ 200, 433, 930, and 2000 mg/kg·BW (ig) (*n* = 8). A subacute toxicology experiment was carried out according to the method of Saheed et al. [27] in doses of EZ 100, 200, and 300 mg/kg·BW (ig) (*n* = 8). EZ was resolved by pure water. Serum was obtained from blood samples after centrifuging at 6000 rpm for 40 min and stored at −80 °C for further analysis.

### 2.5. Animals and Experimental Design

In total, 56 Male C57BL/6J mice (SPF grade, 6 weeks old, 18 ± 2 g) were raised under specific pathogen-free conditions (12 h/12 h light–dark cycle, 22 ± 2 °C) for adaptive fed with a water and low-fat low-sucrose diet for 7 days. All experimental procedures were approved by the Animal Ethics Committee of the Ocean University of China (certificate NO. SYXK20120014). Then the mice were grouped into a control group (N, distilled water), model control group (M, distilled water), and experimental groups. The mice in the N group were fed with low-fat low-sucrose diet (Jiangsu Xietong Pharmaceutical Bio-engineering Co., Ltd., Nanjing, China, #1010009), and the others were fed with a high-sugar and high-fat diet (HSHFD; Jiangsu Xietong Pharmaceutical Bio-engineering Co., Ltd., #XTHSHF60-1) for 90 days to establish the type 2 diabetes model. At the same time, the experiment groups were divided into 5 groups, including the non-ligand Zinc group (NZ, ZnSO_4_, 5 mg/kg·BW, ig), the low-dose EP group (LE, 12.5 mg/kg·BW, ig), the high-dose EP group (HE, 50 mg/kg·BW, ig), the low-dose EZ group (LEZ, 12.5 mg/kg·BW, ig), and the high-dose EZ group (HEZ, 50 mg/kg·BW, ig) (*n* = 8 per group). The low-fat low-sucrose diet (#1010009) consisted of 21.5% of protein, 11.1% of lipid, and 67.4% of carbohydrate. The high-sugar and high-fat diet (#XTHSHF60-1) consisted of 19.6% of protein, 34.3% of lipid, and 33.1% of carbohydrate.

### 2.6. Analysis of Insulin Resistance

After completing the experiment of Section 2.5, the mice were fasted for 10 h. Blood was collected from the mice eyeballs after sacrifice by decapitation under anesthesia (ether), and the serum was separated by centrifuging at 6000 rpm for 30 min, in which the glucose and insulin levels were detected by kits, respectively. HOMA-IR (Homeostasis model assessment-insulin resistance, HOMA-IR) and QUICKI (Quantitative insulin sensitivity check index, QUICKI) were calculated according to Formulas (1) and (2), respectively.
HOMA-IR = fasting blood glucose × serum insulin/22.5(1)
QUICKI = 1/(l g (fasting blood glucose) + l g (serum insulin))(2)

### 2.7. Serum Index Detection

The total cholesterol (TC) and triglyceride (TG) in the serum were determined by TC and TG test kits, respectively. Serum high-density lipoprotein cholesterol (HDL-C), low-density lipoprotein cholesterol (LDL-C), adiponectin (ADPN), resistin, leptin (LEP), tumor necrosis factor α (TNF-α), interleukin 2 (IL-2), and interleukin 6 (IL-6) were determined by corresponding ELISA kits, respectively.

### 2.8. Histological Analysis

The tissues of the liver, epididymal fat, and pancreatic tail of mice were fixed with 4% of paraformaldehyde for 48 h and then embedded in melty paraffin. After solidification, the wax block was made into 4 μm thick slices, which were stained with hematoxylin and eosin (H&E) after dewaxing. An optical microscope was used to observe and photograph the microscopic structures of the tissues.

### 2.9. Quantitative Real-Time PCR

The mRNA relative contents of IRS2 (insulin receptor substrates-2), PI3K (phosphatidylinositol 3-kinase), AKT (serine/threonine kinase, protein kinase B), and GLUT2 (glucose transporter type 2) in the liver of mice were determined by qPCR. The total RNA of each liver and muscle sample was extracted by a HP Total RNA kit and tissue Total RNA kit (OMEGA, Stamford, CT, USA), respectively. The reverse transcription method was used to synthesize the first cDNA strand (5X All-In-One MasterMix; ABM, Vancouver, BC, Canada), then the cDNA strand was amplified by quantitative PCR with SYBR Green (TOYOBO, Osaka, Japan) and gene-specific primers. β-actin was used to normalize the relative expression of each gene which was later analyzed by the 2^−ΔΔCt^ method. The primer sequences are listed in Appendix A.

### 2.10. S RNA Gene Sequence Analysis of Microbiota in Cecum

Mice feeding and fecal acquisition were according to the method described by Chi et al. [28]. The subsequent DNA extraction, construction of cDNA library, and metagenomic sequencing were performed by Majorbio Bio-Pharm Technology Co., Ltd. (Shanghai, China). The amplification of the bacterial 16S rRNA gene V3-4 region, purification of the amplified products, and their quantification were operated by the methods described by Chi et al. [28]. Then, amplicons were pooled in equal amounts, and paired-end 2300 bp sequencing was performed on the Illumina MiSeq platform.

After being cleaned by BWA (Version 0.7.9a), high-quality reads were assembled to contigs using MEGAHIT (Version 1.1.2), and contigs with a length equal or over 300 bp were selected to be the final assembling results.

Image GP (https://www.bic.ac.cn/ImageGP/index.php/Home/Index/Boxplot.html (accessed on 5 April 2023)) was used to analyze alpha diversity, including Shannon, Ace, and Good’s coverage index based on family, genus, and species levels. Principal coordinates analysis (PCoA) was conducted based on the distance matrix between Bray–Curtis samples to compare beta diversities between groups. The relative abundance of the dominant bacteria at family levels was analyzed.

### 2.11. Statistical Analysis

All data were expressed as mean ± SEM. Multiple comparison analyses of differences among all groups were evaluated with the one-way ANOVA and Waller–Duncan test using SPSS20.0 software. The value of *p* < 0.05 was accepted as statistically different.

## 3. Results

### 3.1. Preparation and Optimization of EZ

Optimal reaction conditions of EZ preparation were analyzed by single factor experiment, and the content of Zn(II) was as used as the index. The effects of different pH values on Zn(II) content in the EZ complex is shown in Figure 1a. The increasing pH led to increased Zn(II) content, and the maximum Zn(II) content reached 157.43 mg/g when pH was 5.0 and then the chelation rate began to decrease. The effect of reaction temperature on Zn(II) content is shown in Figure 1b. The results show that the optimal reaction temperature was 50 °C. The influence of reaction time on Zn(II) content in EZ is shown in Figure 1c. The results show that the Zn(II) content increased when the reaction time was prolonged to 90 min and then tended to be stable later. Zn(II) content in EZ was analyzed by the ratio of EP to ZnSO_4_ (Figure 1d). The results show that the Zn(II) content increased with the excessive addition of ZnSO_4_. Based on the principle of rational utilization of resources, the final reaction conditions of EZ preparation were determined as 50 °C, 90 min, pH 5.0, and the mass ratio of EP to ZnSO_4_ was 1:1 (*w*/*w*).

### 3.2. Characterization of EZ

The molecular weight of EP and its Zn(II) complex were 3.31 KDa and 3.10 KDa, respectively (Appendix A). According to the results of the monosaccharide composition of standards (Figure 2a), EZ was composed of rhamnose, glucuronic acid, glucose, and xylose in the molar ratio of 5.29:1.00:1.27:1.03 (Figure 2b). The results of ultraviolet spectrum analysis showed that the absorbance values of EP and EZ were significantly different in the range of 250 nm to 350 nm (Figure 2c), indicating that EP coordinates with zinc ions. The results of FT-IR spectrum analysis of EP and EZ are shown in Figure 2d, and the spectrum was in the range of 400–4000 cm^−1^. O-H stretching vibration resulted in a wide and strong peak that appeared at 3421.68 cm^−1^, owing to the fact that EP molecules could form intramolecular or intermolecular hydrogen bonds easily [29]. When EP was combined with Zn(II), the absorption peak shifted to a higher wavelength of 3423.95 cm^−1^. The absorption bands around 1600 cm^−1^ and 1400 cm^−1^ correspond to the C=O stretching vibration and O-H bending vibration, respectively [30], and these two peaks in EZ moved towards the lower wavelength. The stretching vibration peak of C-O was reported around 1046.84 cm^−1^ [1], which in EZ shifted to a higher wavelength of 1048.30 cm^−1^. These shifted peaks indicate that the EP and Zn(II) might be bound by the formation of an O-Zn bond, and a similar phenomenon has been speculated on and attributed to an electrostatic attraction between the positive charge of zinc ions and the negative charge of polysaccharides [31].

### 3.3. Safety Evaluation

The acute toxicological experiment performed for 18 days showed EZ of less than 2000 mg/kg·BW did not cause acute toxicity to the male and female mice (Appendix A). The subacute toxicological experiments showed that the EZ had no subacute toxicity to male and female mice after a daily gavage of less than 500 mg/kg·BW for 30 days (Appendix A).

### 3.4. Effects of EP and Its Zn(II) Complex on HSHF-Induced Body and Tissue Weights

Weight gain can lead to high glucose and insulin resistance, which can lead to a host of chronic and acute diseases [32]. By establishing a mouse obesity model, HSHF feeding for 90 days led to significant increases in body weight (Figure 3a). The body weight in M group mice was significantly higher than those of other groups (*p* < 0.05). Administrations of Zn(II), EP, and EZ complex all can decrease the body weight of mice, and the body weight of mice in the HEZ group was closer to the normal group, but there is no significant difference among them.

A high dose of EZ can significantly inhibit epididymal and subcutaneous fat accumulation compared with the model group, Zn(II), LE, or LEZ groups (Figure 3c,d) (*p* < 0.05). As shown in Figure 3b, the liver weight was significantly reduced in the HEZ group compared with that in the M group and other experiments group (*p* < 0.05). It can be identified by the H&E staining results of liver tissue (Figure 3e). The liver cells in the N group were intact and orderly in morphology, while the hepatic sinusoids around the central lobule vein in the M group were enlarged. The liver histomorphology in the HEZ group was significantly improved and close to the characteristics of normal liver tissue.

### 3.5. Effects of EP and Its Zn Complex on Glucose and Insulin Resistance

Abnormal glucose is a common symptom of T2DM. Compared with the normal group, the fasting blood glucose level in the M group was significantly increased by 61.27%. After 90 days of intervention, Zn(II), EP, and their complex can decrease the fasting blood glucose level compared with the M group, and the effects of high-dose *E. prolifera* polysaccharide (HE) and its Zn(II) complex is the most significant, but there is no significant difference among other experiment groups (Figure 4a). In response to increased blood sugar, the β cells of the pancreas release a lot of insulin, causing a subfigures compensatory rise in insulin, which can lead to insulin resistance. As shown in Figure 4b, the accumulation of insulin could be inhibited by Zn(II), EP, and their complex administration, in which the effect of HEZ is the most significant (*p* < 0.05).

Insulin resistance (IR) and β cell dysfunction, such as deficiency in insulin secretion, are critical factors in the development of Type 2 diabetes [33]. High HOMA-IR indicates a high IR degree. QUICKI is used to assess insulin sensitivity [34]. As shown in Figure 4c,d, the HOMA-IR and QUICKI index in the M group had significantly higher and lower, respectively, signifying impaired β-cell function and IR formation. Meanwhile EP, Zn(II), and their complex administration had significantly inhibited the increasing HOMA-IR index and increased QUICKI index (vs. the model group), especially for HEZ, indicating that EZ could effectively inhibit the degree of insulin resistance and enhance insulin sensitivity.

The effects of EP and EZ on pancreatic tissue histology are shown in Figure 4e and Appendix A. The pancreatic tissue of mice in the N group contained abundant islet cells, with relatively regular arrangement, a relatively complete cell morphology, and more cytoplasm compared to that of the M group. Comparing with the N group (Figure 4e-N), pancreatic tissue in the M group was significantly atrophied, irregularly shaped around the cells, and became smaller (Figure 4e-M). EP and its zinc complex administration in mice significantly improved the atrophied pancreatic tissue, suggesting that Zn(II) and its polysaccharide complex improved the pancreatic tissue function and the effect of HEZ was the most significant.

### 3.6. Effect of EP and Its Zn Complex on Lipid-Related Indexes in Mice

IR is the main cause of dyslipidemia, and lipid metabolism disorders are usually accompanied by T2DM. Controlling blood lipid levels can prevent and delay the occurrence of T2DM and its related complications [35]. Compared with the N group, the serum levels of TC, TG, and LDL–C in the M group were significantly increased (Figure 5a,b,d), while the level of HDL–C was significantly decreased (Figure 5c). EP and its Zn(II) complex intake decreased the serum TC, TG, and LDL–C and increased the HDL–C level, in which the effect of HEZ was the most significant. The results indicate that EZ can restore the disordered lipid metabolism caused by high-sugar and high-fat feeding, regulate the blood lipid level, and promote the formation of normal blood lipid levels.

### 3.7. Effect of EP and Its Zn Complex on Adipokines and Inflammatory Factors in Mice

Abnormal adipocytokines are also associated with insulin resistance [36]. Adiponectin (ADPN) secreted by adipocytes plays a key role in insulin-sensitive tissues. Compared with the normal group, the serum ADPN level of mice in the M group decreased and insulin resistance increased, while Zn(II), EP, and their complex significantly reversed the decrease of ADPN, in which the effect of HEZ was the most significant. There was no significant differences among the NZ, LE, HE, and LEZ groups (Figure 6a).

The serum resistin of mice in the M group increased, which promoted insulin resistance and caused inflammation. While high dose of EP and its Zn(II) complex decreased the resistin contents from 11.42 ng/mL of the M group to 8.50 ng/mL and 6.84 ng/mL, respectively (Figure 6b). Compared with the normal group, the leptin content in the M group was significantly increased, indicating that obesity led to the compensatory increase of leptin in mice, leading to leptin resistance. EP and its Zn(II) complex supplementation could significantly reduce the leptin content, in which the leptin content in the high-concentration *E. prolifera* polysaccharide (HE) group was reduced by 34.79% (Figure 6c). This is consistent with the results of HE staining of adipose tissue of mouse epididymis (Figure 6g). The epididymal adipocytes in the N group were small, while those in the M group were hypertrophic. It indicates that high-glucose and hig-fat feeding leads to obesity and promotes the accumulation of fat. EZ supplementation effectively inhibited the growth of adipocytes and inhibit the accumulation of fat.

The inflammation caused by insulin resistance can be identified by the increasing level of the serums TNF-α, IL-2, and IL-6 in the M group, indicating the inflammatory response is caused by obesity (Figure 6d–f). EP and its Zn(II) complex could decrease the inflammation levels (*p* < 0.05), but there is no significant difference among Zn(II), EP, and their complex groups for IL-2 and IL-6 index.

### 3.8. Effect of EP and Its Zn Complex on mRNA Expression of IRS-2, PI3K, AKT, and GLUT2 Genes in Liver

The effect of EP and its Zn complex on the mRNA expression of IRS-2, PI3K, AKT, and GLUT2 in the hepatic tissues of each group is shown in Figure 7a–d. The relative mRNA gene levels of IRS-2, PI3K, AKT, and GLUT2 were significantly downregulated by 48.60%, 79.07%, 89.22%, and 100% (*p* < 0.05) in the M group compared with the N group. The administration of NZ and HEZ up-regulated IRS-2 mRNA levels significantly in hepatic tissues, and the improvement effect of HEZ was the most significant (Figure 7a). NZ, EP, and their complex could restore the mRNA levels of PI3K, AKT and GLUT2 in different extent compared with M group (*p* < 0.05). The results suggest that long-term administration of EP and its zinc complex activated the glucose metabolism-related gene signaling pathway (IRS/PI3K/AKT) and up-regulated the expression of GLUT2 gene, thus decreased HOMA-IR and increased the absorption of glucose by the liver, thereby improving the symptoms of type 2 diabetes.

### 3.9. Effect of EP and Its Zn Complex on HSHF-Induced Species Composition of Gut Microbiota

Alpha diversity analysis based on the Shannon index at the family, genus, and species levels (Figure 8(a1)) reveals that HEZ could significantly improve the Shannon index compared with that of the M group, indicating that HEZ administration could improve the intestinal microbial community diversity in mice. HEZ also significantly improved the ACE (abundance-based coverage estimator, ACE) index compared with that of the M and N groups (Figure 8(a2)), indicating that HEZ administration could help increase the intestinal microbial community richness in mice. The Good’s coverage values were all larger than 0.999 (Figure 8(a3)), indicating that the sequencing results represent the true situation of the microorganisms in the samples.

Principal coordinates analysis (PCoA) at the OTU level suggested that HSHF feeding significantly altered the distribution of intestinal microbiota in the M group (Figure 8b), while the HEZ administration group exhibited similar intestinal microbiota compositions to the N group. Analysis of the similarities among the different groups based on Bray– Curtis distance demonstrated that the differences among the groups were greater than those within each group (Figure 8c), indicating that the experimental groupings are meaningful and HEZ administration has a significant positive effect on the gut microbiota distribution.

The community abundance histogram of the gut microbiota at the family level is presented in Figure 8d. Compared with the N group, the abundances of *Staphylococcaceae*, *Planococcaceae*, *Moracellaceae*, *Muribaculaceae*, *Aerococcaceae*, *Lacrobacillaceae*, *Enterococcaceae*, and *Micrococcaceae* were significantly decreased in the M group, whereas those of *Clostridiaceae*, *Bifidobacteriaceae*, *Oscillospiraceae*, and *Marinifilaceae* were significantly decreased. Following treatment with HEZ, the abundance of *Staphylococcaceae*, *Planococcaceae*, *Moracellaceae*, *Muribaculaceae*, *Aerococcaceae*, *Lacrobacillaceae*, *Enterococcaceae*, and *Micrococcaceae* as the resident intestinal bacteria increased, whereas those of *Faecalibaculum* and *Clostridiaceae* decreased. *Lacrobacillaceae* and *Bifidobocteriaceae* are also intestinal probiotics. All the results suggest that HEZ administration could improve the intestinal microbiota closer to the normal level in species and abundance.

## 4. Discussion

Diabetes is a kind of chronic metabolic disease, and the most direct manifestation is the rise of blood sugar in the body. T2DM has the largest number of patients in the world. T2DM induces dysregulation of glucose–lipid metabolism and eventually develops complications such as hyperlipidemia, liver damage, inflammation, and oxidative stress. Zn(II) deficiency is common in people with diabetes. The effect of Zn(II) in regulating sugar metabolism has been previously reported [37,38]. Supplementation of Zn in an appropriate way has become an important way to develop hypoglycemic drugs with high bioavailability and low toxicity [38,39]. Several studies on polysaccharides or their oligo polymers, such as carrageenan, ulvan, and fucoidan oligosaccharides, have also shown their positive effects on obesity and related diseases [40,41]. Additionally, EP was reported to possess excellent chelating properties [22]. Therefore, *E. prolifera* polysaccharide was employed to chelate with Zn(II) to investigate its hypoglycemic activity in the present research. T2DM has been reported to have a higher insulin level in serum and is characterized by insulin resistance, which is caused by the high-fat diet in STZ-induced diabetic rats [42]. In our research, the HSHF diet fed led to a significant increase in body weight, liver weight, blood lipid levels, fasting blood glucose, serum insulin, and HOMA-IR index. Increasing fasting blood glucose levels induced the release of more insulin by pancreas islet β cells, resulting in the compensatory increase of insulin, leading to the formation of insulin resistance and causing pancreas islet atrophy. EP and its Zn(II) complex administration could inhibit the compensatory increase of insulin and decrease the release of insulin and HOMA-IR, in which the effect of HEZ is the most significant. The antidiabetic activity of EP and Zn(II) complex is consistent with the report of D. opposite Thunb polysaccharide-zinc complex by Zhang Y. et al. [1].

The pancreas is the only organ in the body that secretes insulin. The decrease of insulin secretion leads to an increase in blood sugar, and the atrophy of pancreatic tissue greatly affects the biological function of the pancreas [43]. HEZ administration repaired the pancreas atrophy induced by HSHF diet feeding, which can be identified by histological results in our research.

On the other hand, many studies have shown that diabetes is related to lipid metabolism [44]. In the present research, HSHF not only induced the formation of hyperglycemia but also increased the levels of serums TC, TG, LDL-C, resistin, and LEP, and decreased the level of ADPN. ADPN and resistin can regulate insulin sensitivity and reduce glucose output and insulin resistance [45,46,47]. HEZ could significantly improve the glucose level by regulating the levels of ADPN and resistin. At the same time, ADPN and resistin were also reported to associate with inflammation, and the increasing ADPN and decreased resistin induced by HEZ reduced inflammatory cytokines levels, such as IL-2, IL-6, and TNF-α in HSHF mice.

In terms of mechanism, PI3K/AKT is considered the key regulator in the insulin signaling pathway, and PI3K/AKT pathway inhibition will frequently cause hepatic insulin resistance [48,49]. IRS-2 can mediate the translocation of the insulin-responsive glucose transporter, while insulin-mediated downregulation of PI3K/AKT signal transduction is associated with the inhibition of glucose transduction transport and reduced glucose uptake [50]. GLUT2 is the major glucose transporter in liver cells and is necessary for liver glucose uptake. Therefore, the impact of EP and its Zn complex on mRNA expression of IRS-2, PI3K, AKT, and GLUT2 in the hepatic tissues of each group was investigated in present research. The downregulated PI3K, AKT, and GLUT2 mRNA levels indicate that HSHF feeding could induce hepatic insulin resistance, while *E. prolifera* polysaccharide and its Zn(II) complex significantly decreased insulin resistance and improved glucose transport, thereby showing hypoglycemic activity, in which the modulatory effect of Zn(II) on insulin signaling is possibly associated with phosphorylation of the insulin receptor IRS-2, activation of protein kinase B (Akt) signaling pathway, and increase of GLUT-2 expression and translocation.

On the other hand, according to our research data, Zn(II), *E. prolifera* polysaccharide and their complexes all showed hypoglycemic activity to different extents, in which the regulating ability of the HEZ complex was more significant, especially compared with the Zn(II), EP, and LEZ group on liver weight, epididymal fat weight, subcutaneous fat weight, fast blood glucose, serum insulin, and lipid indexes. We speculate that the high dose of the complex would be a reservoir of Zn, allowing for its slow release followed by the process of the complex being degraded in the intestine and prolonging the effective Zn(II) release time. The exact mechanism needs to be further investigated.

A healthy gut microbiota can improve human metabolism, regulating immunity, anti-inflammation, antioxidant, and anti-aging during the body’s growth, development, and aging process [51]. Gut microbiota also plays an important role in the pathophysiology of various complex diseases [15,52,53]. Environmental or genetic factors could impair intestinal integrity in obese animals and then induce gut dysbiosis [54], leading to the release of the endotoxin lipopolysaccharide (LPS) from intestinal Gram-negative bacteria into the bloodstream and subsequently leading to metabolic inflammation and insulin resistance in obese mice [55], which has been identified by the increasing levels of IL-2 and IL-6 in the M group in the present research. Moreover, it can change the bacterial community, including the number of species present, numerical composition, and bacterial diversity [56], which has been identified through the loss of community diversity indexed by Shannon, and the loss of community richness indexed by ACE in the present research. HEZ administration improved the indexes mentioned above and improved the dominant intestinal microbiota closer to the normal level in species and abundance, which was identified by principal coordinates analysis, ANOSIM analysis, and species composition analysis of mouse gut microbiota at the family level.

## 5. Conclusions

In this study, *Enteromorpha prolifera* polysaccharide(EP)–Zn(II) (EZ) complex was obtained by chelating EP with Zn(II), preliminary characterized and investigated in hypoglycemic ability. The pharmacological study revealed that EP and its complex had a potent hypoglycemic effect on high-sugar and high-fat-induced diabetic mice by inhibiting insulin resistance, improving dyslipidemia, decreasing inflammatory factors, repairing pancreas damage, as well as activating the IRS/PI3K/AKT signaling pathway and regulating GLUT2 gene expression, in which the effects of HEZ were the most significant compared with M, non-ligand Zn(II), EP, and LEZ groups. It suggests that a high dose of the polysaccharide–Zn(II) complex may provide more effective Zn(II) as a supplement for pancreas islet by prolonging the release of Zn(II) possibly. Moreover, our research indicates that HEZ enhances the abundance of dominant species in intestinal microbiota distribution. Thus, EP and its complex could be considered as a potential candidate for developing an ingredient of functional foods for Zn(II) supplements with hypoglycemic activity.

## Figures and Tables

**Figure 1 foods-12-02854-f001:**
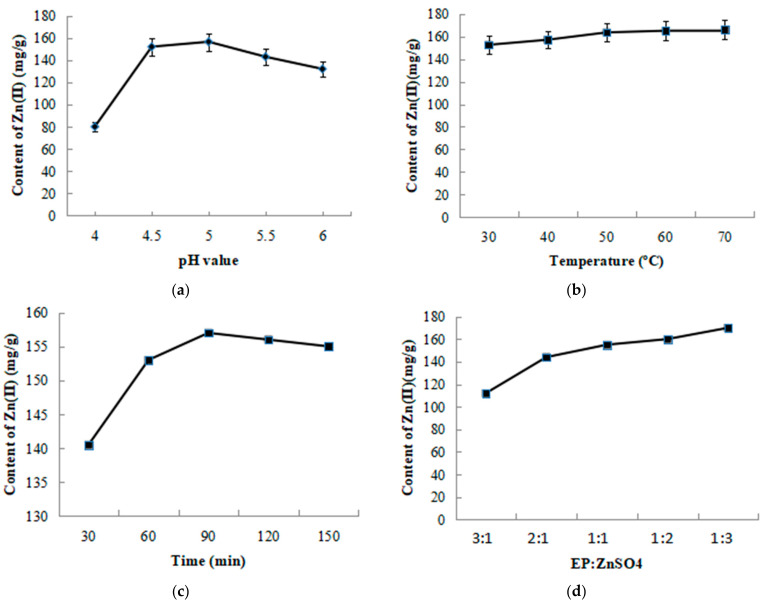
Analysis of single-factor experiments. (**a**) pH, (**b**) temperature, (**c**) time, and (**d**) mass ratio of EP to ZnSO_4_.

**Figure 2 foods-12-02854-f002:**
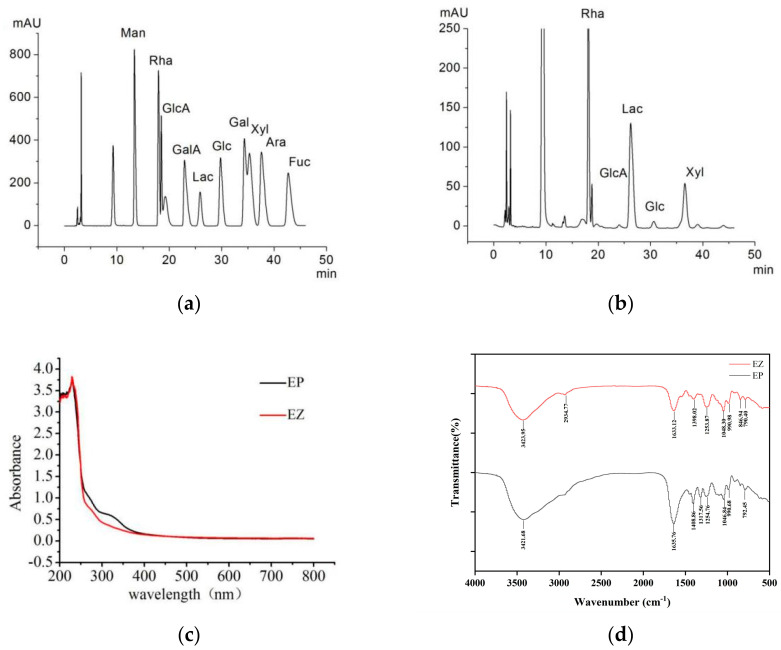
Characteristic analysis of EZ and EP. (**a**,**b**) Monosaccharide composition of monosaccharide standards (**a**) and EZ (**b**), (**c**) UV spectrum scanning, (**d**) FT–IR spectrums.

**Figure 3 foods-12-02854-f003:**
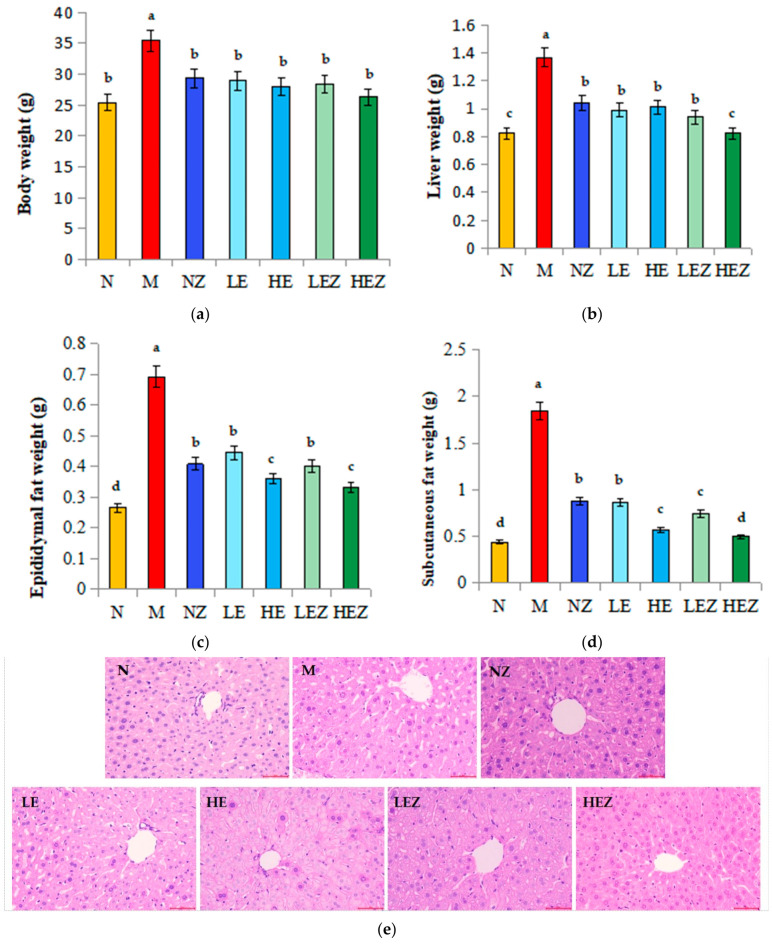
Effects of EZ on body weight and fat weight in HSHF diet-fed mice (*n* = 8). N, normal; M, model; NZ, non-ligand Zn(II); LE, low dose of *E. prolifera* polysaccharide; HE, high dose of *E. prolifera* polysaccharide; LEZ, low dose of *E. prolifera* polysaccharide–Zn(II) complex; HEZ, high dose of *E. prolifera* polysaccharide–Zn(II) complex. (**a**) Body weight; (**b**) liver weight; (**c**) epididymal fat weight; (**d**) subcutaneous fat weight; (**e**) H&E staining of mouse liver tissue, 20×. The letters in subfigure (**a**–**d**) express the averages in multiple comparison analyses of differences, a > b > c > d, *p* < 0.05.

**Figure 4 foods-12-02854-f004:**
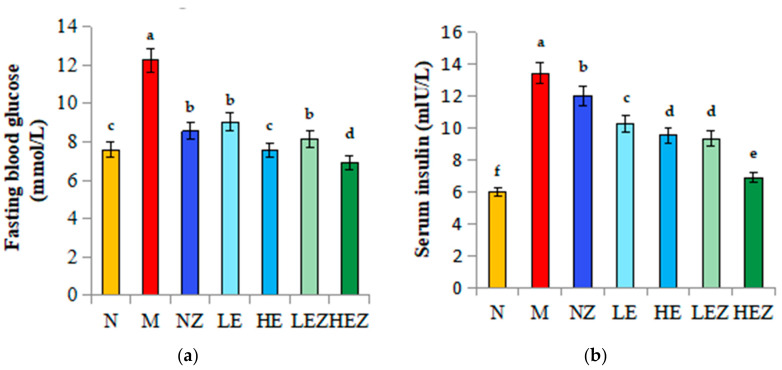
Effects of EZ on fasting blood glucose, serum insulin, and insulin resistance indexes in high–sugar and high–fat diet-fed mice (*n* = 8). N, normal; M, model; NZ, non–ligand Zn(II); LE, low dose of *E. prolifera* polysaccharide; HE, high dose of *E. prolifera* polysaccharide; LEZ, low dose of *E. prolifera* polysaccharide–Zn(II) complex; HEZ, high dose of *E. prolifera* polysaccharide–Zn(II) complex. (**a**) Fasting blood glucose; (**b**) serum insulin; (**c**) HOMA-IR; (**d**) QUICKI; (**e**) H&E staining of mouse pancreas, 20×. The letters in subfigure (**a**–**d**) express the averages in multiple comparison analyses of differences, a > b > c > d > e > f, *p* < 0.05. The arrow in subfigure (**e**) indicates the position of pancreas.

**Figure 5 foods-12-02854-f005:**
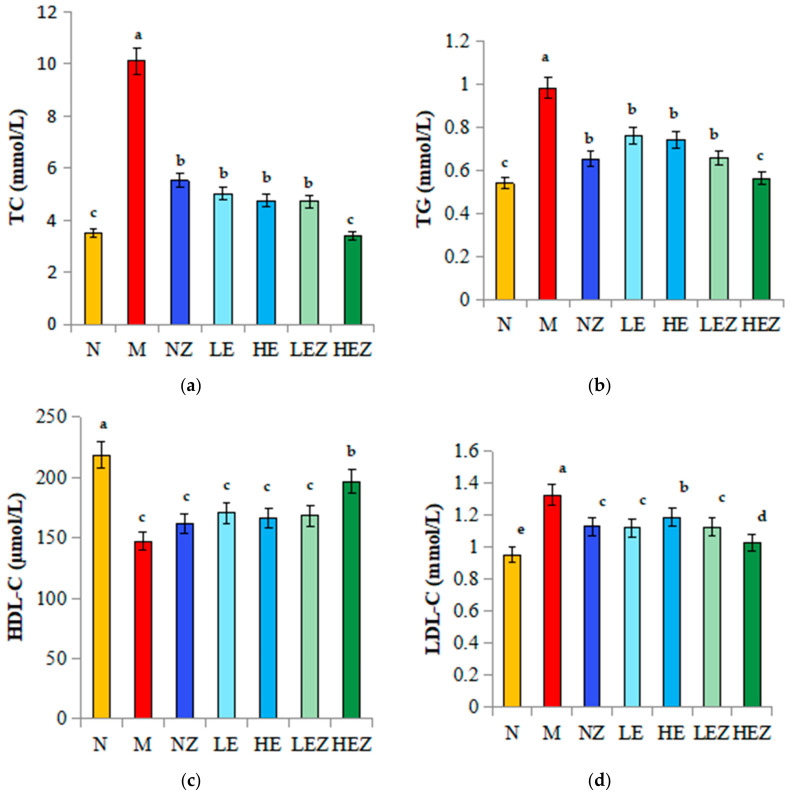
Effects of EZ on blood lipid levels in high-sugar and high-fat diet-fed mice (*n* = 8). N, normal; M, model; NZ, non-ligand Zn(II); LE, low dose of *E. prolifera* polysaccharide; HE, high dose of *E. prolifera* polysaccharide; LEZ, low dose of *E. prolifera* polysaccharide–Zn(II) complex; HEZ, high dose of *E. prolifera* polysaccharide–Zn(II) complex. (**a**) TC; (**b**) TG; (**c**) HDL–C; (**d**) LDL–C. The letters in subfigure (**a**–**d**) express the averages in multiple comparison analyses of differences, a > b > c > d > e, *p* < 0.05.

**Figure 6 foods-12-02854-f006:**
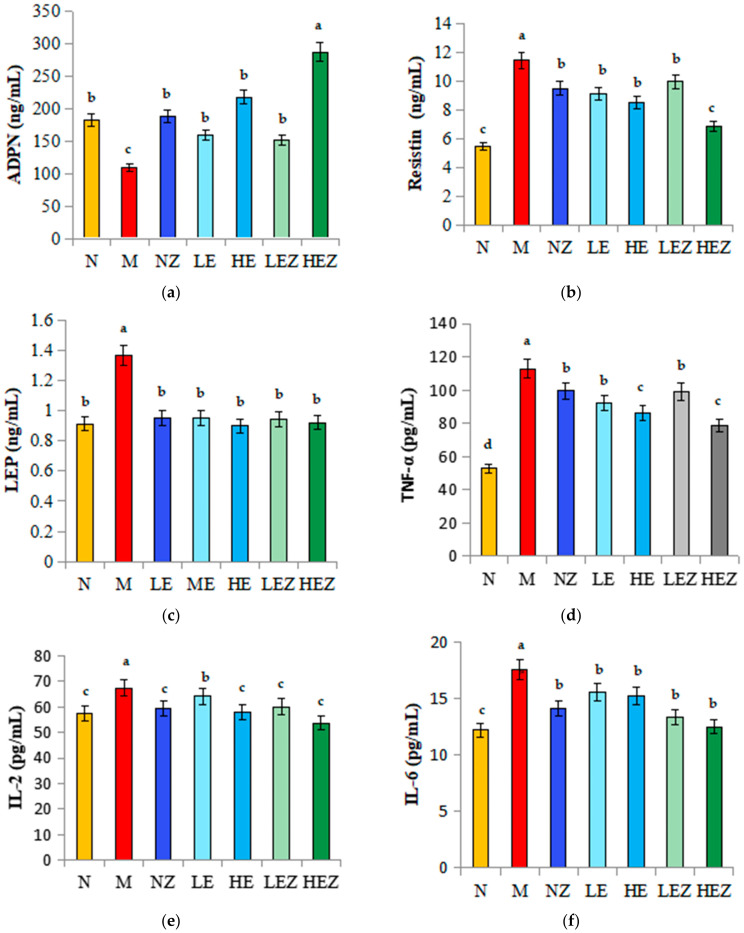
The effect of EZ on adipocytokines in high-sugar and high-fat diet-fed mice (*n* = 8). N, normal; M, model; NZ, non-ligand Zn(II); LE, low dose of *E. prolifera* polysaccharide; HE, high dose of *E. prolifera* polysaccharide; LEZ, low dose of *E. prolifera* polysaccharide–Zn(II) complex; HEZ, high dose of *E. prolifera* polysaccharide–Zn(II) complex. (**a**) Adiponectin; (**b**) Resistin; (**c**) Leptin (LEP); (**d**) TNF–α; (**e**) IL–2; (**f**) IL–6; (**g**) H&E staining of mouse epididymal fat tissue, 20×. The letters in subfigure (**a**–**f**) express the averages in multiple comparison analyses of differences, a > b > c, *p* < 0.05.

**Figure 7 foods-12-02854-f007:**
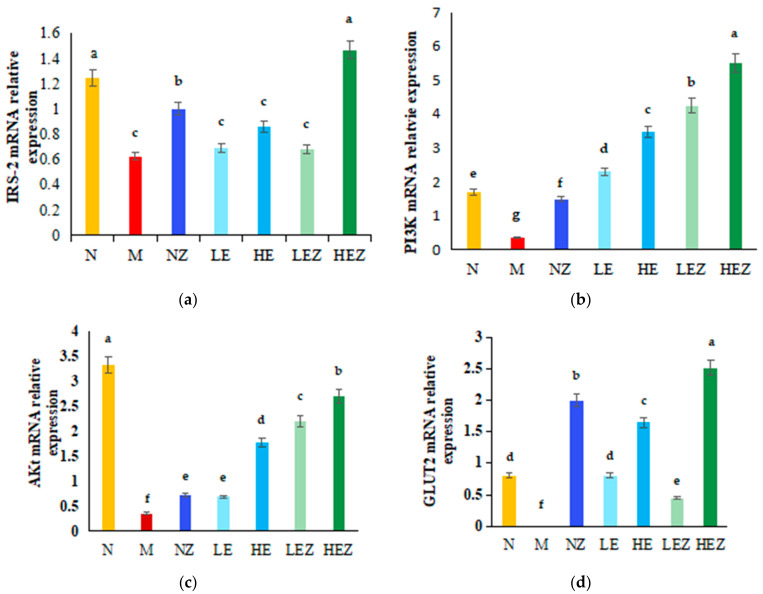
Effect of EZ on the relative mRNA content of IRS-2 (**a**), PI3K (**b**), AKT (**c**), and GLUT2 (**d**) in the liver of mice. N, normal; M, model; NZ, non-ligand Zn(II); LE, low dose of *E. prolifera* polysaccharide; HE, high dose of *E. prolifera* polysaccharide; LEZ, low dose of *E. prolifera* polysaccharide–Zn(II) complex; HEZ, high dose of *E. prolifera* polysaccharide–Zn(II) complex. The letters in subfigure (**a**–**d**) express the averages in multiple comparison analyses of differences, a > b > c > d > e > f > g, *p* < 0.05.

**Figure 8 foods-12-02854-f008:**
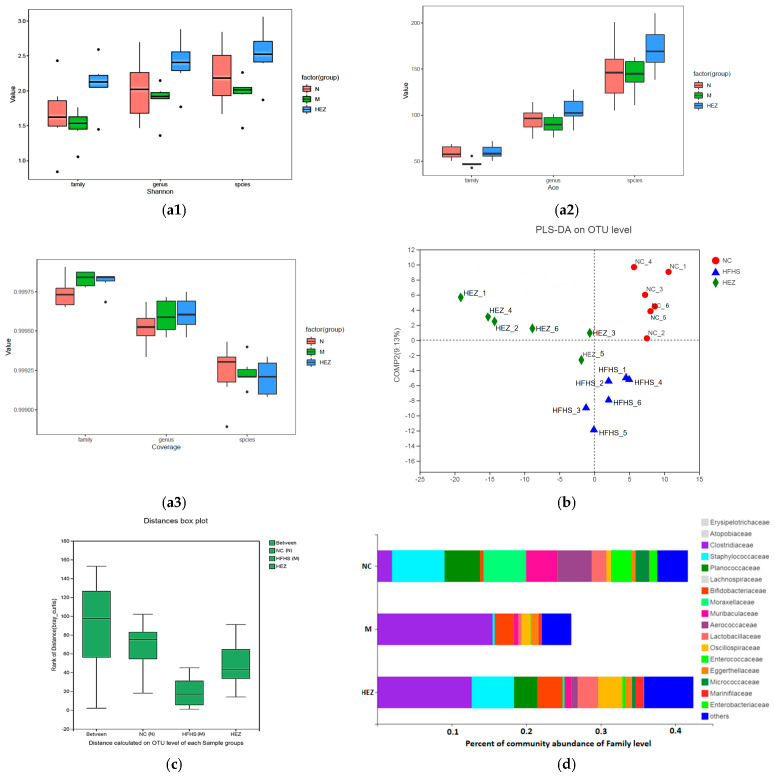
Effect of treatment with the polysaccharide–Zn(II) complex (EZ) on species composition of gut microbiota. (**a1**–**a3**) Analysis of α diversity based on Shannon (**a1**), Ace (**a2**) and coverage (**a3**); (**b**) principal coordinates analysis; (**c**) ANOSIM analysis between different group; (**d**) analysis of species composition of mouse intestinal flora on family level.

## Data Availability

Data is contained within the article or Appendix A.

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
