# Peer review of "Regulatory Effects Mediated by Enteromorpha prolifera Polysaccharide and Its Zn(II) Complex on Hypoglycemic Activity in High-Sugar High-Fat Diet-Fed Mice"

_foods, 2023, doi:10.3390/foods12152854_

Round 1
Reviewer 1 Report
1- Abstract section must be include the main task of aim of this article.
2- Introduction section must be contain an updated recent references.
3- Authors must be added a single paragraph in the last of introduction section contain the important of the study.
4- Figure of infrared spectra must be represented.
5- Conclusion section need to be rewrite to become more interesting.
good
Author Response
Pleas see the atttachment.

Reviewer 2 Report
The manuscript by Li et al aims to establish the anti-diabetic effect of Enteromorpha prolifera polysaccharide-zinc complex in obese mice fed high-fat high-sucrose diet. The manuscript seems interesting, but it has one major flaw considering that the claimed higher preventive effect of high EP-zinc (HEZ) complex is not supported by the statistical analysis. In many figures/analyses, HEZ has a similar and not higher preventive effect compared to EP or zinc. Furthermore, the manuscript needs to be extensively edited in order to improve English language and grammar, to eliminate mistyping errors and inconsistencies, to correct incomplete sentences and wrong verb tenses (see specific comments).
1) Abstract Line 12. Detailed…
2) Lines 17-18. Change intestinal flora into microbiota. Specify dominant species
3) Lines 26-28; 29-33; 45-49; 73-74. English must be improved. Vital theoretical?
4) Line 36. Zinc or Zn. Be consistent. What Zn (II) is? Please, define
5) Lines 58-59. Based on the basic conditions..? What does it mean?
6) Line 73. SPF grade ICR mouse? Please, specify. Strain? Number of animals? only one mouse?
7) Line 82. Why SPF? How many animals?
8) Line 86. After being adaptive fed? Please, rephrase
9) Lines 88 and 90. Composition of diets must be reported
10) Line 96. After completing the experiment of 2.5 of what?… something is missing here.
11) Line 106. Specify which kits
12) Lines 108, 122. Spaces
13) Line 113. Define HE here and elsewhere (line 315). HE is also high dose EP group…
14) Line 124. EP?
15) Line 118. Add the list of primers. Reverse transcriptase using polyT primer or random primers?
16) Lines 147-153. Indicate values of the fixed variables in Fig 1a,b,c and d.
17) Paragraph 3.3 Data of safety evaluation should be shown as supplementary material.
18) Bw or BW? Be consistent
19) Line 203.. to a host of chronic diseases?
20) Fig. 3a. Change the graph on body weight gain showing initial weight and progression through 90 days including statistical analysis. Fig. 3b-c. Is HEZ significantly different compared to LEZ or LE, HE? Apparently not. Additional statistics is needed here and Fig. 4a-d, Fig. 5a-d, Fig. 6a-f, Fig. 7a-d
21) Lines 206-209. English must be improved.
22) Line 214. Orderly in morphology
23) Line 241. Italics
24) Line 243. Beta or b in symbol. Be consistent here and throughout the manuscript.
25) Line 248. Inducements seem an inappropriate term
26) Line 258. … more cytoplasm compare to?
27) Line 262. A quantification of pancreatic cells would be useful.
28) Line 289.. intaking.. meaning?
29) LDL-c or LDL-C?
30) Lines 308-311. Sentences incomplete
31) Line 315. Which is the subject?
32) Lines 317-319. Improve English. Wrong verb tenses.
33) Line 320. The inflammatory? Something is missing here.
34) Fig. 8d. Legens and axis are too tiny. Please, enlarge.
35) Line 429. So… is colloquial. Remove it.
It must be imrpoved. Please, see comments sent to the authors.
Reviewer 3 Report
The manuscript presents interesting results demonstrating the hypoglicemic action of the zinc complex with polysaccharide from Enteromorpha prolifera and its beneficial effects on serum lipid parameters in high-fat diet mouse. The results, especially those obtained for the higher dose of the complex are convincing. The study demonstrates also that the complex activated the IRS/PI3/AKT signaling pathway, induced GLUT2 expression and affected intestinal microbiota.
Remarks
Some diagrams (Figures 4,5 and 7) should be re-drawn more carefully, with more exact positions of markers of statistical significance.
The differences of only M groups with respect to the N group are indicated; other groups are compared with the M group only. It makes sense but perhaps comparison also with the N group would allow to see the extant of normalization with respect to the group fed normal diet.
What was the molecular weight of the polysaccharide?
How was the polysaccharide and its Zn complex administered? In suspension? (if so, what was the concentration and thus the volume introduced?
What can be the mechanism of action of the complex? Does it simply act as a reservoir of Zn, allowing for its slow release or is there something more? This question could be addressed under Discussion.
Reviewer 4 Report
The manuscript by Li et al. describes the regulatory effects of Enteromorpha prolifera polysaccharide (EP) and its zinc complex (EZ) on hypoglycemic activity in high-sugar high-fat diet-fed mice.
In my opinion, the manuscript presents interesting data. However, the main problem is the presentation of data. In particular the caption to Figures should be implemented and the statistical analysis is not complete in the actual form. Moreover, several parts of the manuscript (Introduction and Discussion in particular) require implementation.
Specific comments:
- Lines 28-29: The dietary uses of Enteromorpha prolifera should be describe in the introduction section.
- Lines 44: Please change “Enteromorpha prolifera” to “E. prolifera”.
- Lines 45-47: The aim of the research and the model/techniques used to obtain it should be better introduced/indicated in the Introduction section.
- Data on safety evaluation of EZ in comparison to Z and EP should be reported in the manuscript.
- Lines 111-113: The techniques used to prepare and visualize histological tissue sections should be better described.
- Lines 113: Please specify HE.
- Lines 241 and 443: “Enteromorpha prolifera” should be changed to “E. prolifera” and written in Italics.
- Lines 436, 437, 442, 450: The indication of Fig.4b, Fig.4c, Fig.4e, and Fig.4e should be deleted from the Discussion section.
- Please indicate in the legend to Figures 3, 4, 5, 6 and 7 the explanation for each sample group (M, N, NZ, LE, HE, LEZ, HEZ).
- The explanation of the statistical analysis of differences should be added in the caption to Figure 7. Moreover the panels (c) and (d) of Figure 7 are not well-organized.
- The statistical analysis of differences should be also performed among sample groups NZ, LE, HE, LEZ, and HEZ and reported in the corresponding Figures. The indication of statistically significant differences (when present) for EZ versus NZ and EP groups in Figures 3, 4, 5, 6 and 7 is essential to justify the use of the EP-zinc (EZ) complex instead of NZ or EP alone.
- Figure 8 is too complex and should be simplified/reorganized. Moreover, the caption should be improved to better describe all reported experiments.
- The Discussion section is too short and should be implemented.
- Several typographical errors are present in the text. Please revise it.
Minor editing of English language are required.
Round 2
Reviewer 4 Report
The manuscript has been greatly improved and all suggested corrections have been made.
However, in my opinion the explanation of the statistical analysis of differences (multiple comparison analysis of differences among all groups and noted using the letters a, b, c…, p < 0.05) should be added in the caption to all figures (when statistical analysis is present).